# Using Intervention Mapping to Develop an Education and Career Support Service for Adolescents and Young Adults Diagnosed with Cancer: Identification of the Contextual Factors That Influence Participation in Education and Employment

**DOI:** 10.3390/cancers14194590

**Published:** 2022-09-22

**Authors:** Esther L. Davis, Kristina S. Clarke, Pandora Patterson, Jennifer Cohen

**Affiliations:** 1Canteen, Sydney, NSW 2042, Australia; 2Faculty of Medicine and Health, The University of Sydney, Sydney, NSW 2006, Australia; 3School of Clinical Medicine, UNSW Medicine & Health, University of NSW, Sydney, NSW 2031, Australia

**Keywords:** adolescent, young adult, neoplasms, intervention mapping, needs assessment, education, employment, return to work, quality of life

## Abstract

**Simple Summary:**

Quality of life for adolescents and young adults (AYAs) is driven by their participation in education and employment. This participation is disrupted for AYAs diagnosed with cancer. There is limited available information on factors that impact participation in education and employment, as well as limited access to evidence-based services. This paper presents the results from the first step in developing an education and career support service for AYAs diagnosed with cancer using Intervention Mapping. Information was collected and combined from a literature review, survey of AYAs, and feedback from a planning group. Factors found to impact AYAs’ participation in education or employment were categorised under AYA behaviours, environmental conditions, health and demographic factors, and internal factors. This information will guide the development of an education and career support service for AYAs diagnosed with cancer, with the aim of improving quality of life.

**Abstract:**

Adolescents and young adults (AYAs) diagnosed with cancer experience disrupted engagement in education and employment, which can have profound and long-term impacts on their quality of life. It is therefore vital to offer AYAs access to tailored, evidence-based services to help them to achieve their education and employment goals. However, few such services exist for this population. This paper presents the results from the first step in developing an education and career support service for AYAs diagnosed with cancer using Intervention Mapping. This first step involved developing a logic model that describes the influences of health and demographic factors, individual determinants, behaviours, and environmental conditions on AYA participation in education or employment. The logic model was developed by integrating data from an integrative literature review; cross-sectional survey of AYA clients of a community-based organisation; and feedback from a planning group of stakeholders. It is a valuable framework that will be used to direct the focus of the education and career support service for AYAs diagnosed with cancer. More broadly, the logic model has implications for guiding clinical, service, research, and policy improvements for AYA education, employment, and career support, with the aim of improving AYA quality of life.

## 1. Introduction

Adolescence and young adulthood are key life-stages for advancing toward independence and cultivating future aspirations regarding education and career [1,2]. A cancer diagnosis can result in long-term disruption to development for adolescents and young adults (AYAs) and compound already significant challenges to their health and wellbeing [3,4]. AYAs with a cancer diagnosis can experience physical, psychosocial, and behavioural late effects, which potentially disrupt the developmental milestones typical of this age, such as completing high school, pursuing further education or career goals, and becoming financially and socially independent from their family [1,5,6]. 

For AYAs of school-age, cancer and its treatment disrupt school attendance up to three years post-diagnosis [1,7]. Disrupted educational attendance is associated with reduced educational attainment, impacting employment outcomes and career development [5]. Young adult cancer survivors are more likely to be unemployed or underemployed than their same-age peers [1,8] and up to 50% of AYAs identified ongoing difficulties with attaining their prior education or employment goals two years post-diagnosis [9]. As education and employment are social determinants of health [10,11], disruptions to participation and engagement can have a significant impact on the future health and wellbeing of AYAs [12,13,14]. Being under- or unemployed is associated with reduced economic productivity, social participation, and financial insecurity, as well as poor physical and psychological health and quality of life [1,5,8]. 

Despite the demonstration of poor outcomes, many AYAs with cancer report an unmet need for support with acquiring and maintaining education and employment [1,5,9,15]. Young people with a cancer diagnosis report not feeling supported to cope with the changes to their education and/or employment during treatment [16] and difficulty accessing an education and career advisor [5]. AYAs with cancer describe specific education and career support needs, including a need for information about the effects of cancer and its treatment on participation in education and employment [1], personal consultation on career development [5], and job readiness and work skills training [8]. Although support from peers [17] and parents [16] plays an important role in educational or career attainment, AYAs described feeling better supported when they received formal support from education and career professionals, rather than informal support from parents or friends [16]. Despite the evidence for the efficacy of formal education and career support, many young people report being unable to access this type of specialised professional support [5,9]. 

### 1.1. Education and Career Support Services

Programs to facilitate transition to higher education and into the workforce can improve health outcomes [11], yet there are few evidence-based interventions developed to meet the specific needs of AYAs diagnosed with cancer [1]. Two conceptual frameworks have been developed to guide the design and delivery of education and career interventions for AYA with cancer. The Illinois Work and Wellbeing Model (IWWM) was developed as a framework of career development for young adult cancer survivors [8]. It highlights the potential for an intervention to impact the bi-directional relationships between individual, environmental, and cancer-related contextual factors; career development processes of being aware of, attaining, and keeping a job; and participation in society [8]. Doidge and colleagues adapted the Canadian Model of Occupational Performance and Engagement (CMOP-E) for AYAs with cancer [9]. This model identifies environmental (e.g., lack of peer support), personal (e.g., poor physical, cognitive, and affective functioning), spiritual (e.g., loss of identity), and productivity (e.g., absenteeism) challenges to engaging in productive occupations. The challenges identified in the model have been applied to the development and evaluation of an education and vocation support service within an AYA cancer treatment centre [5], with promising results in supporting AYAs to remain engaged in work or study throughout their cancer trajectory [5]. Given that health interventions with a theory base have been shown as more effective than those without a theory base [18], the examples described above demonstrate the value of conceptual frameworks in guiding intervention development and ensuring fit with the targeted population and setting. 

### 1.2. Intervention Mapping

The systematic Intervention Mapping (IM) framework provides a methodical six-step protocol for program planning, which spans assessment of a target population’s needs and the development, implementation, and evaluation of a theory- and evidence-informed intervention that addresses these needs [19,20]. IM adds value to intervention development through its integration of theory with evidence, consideration of individual and environmental contributors to a public health problem, and participatory involvement at each step of the process to ensure fit between population needs and the intervention context [19]. IM has been successfully used to develop effective interventions in the cancer field [21,22,23] and among AYAs [24,25], indicating its utility as a protocol to develop an education and career intervention for AYAs with a cancer history. 

The first step in IM aims to define the health problem to be addressed, identify the behaviours and environmental conditions that contribute to the health problem, and further identity the determinants of those behaviours and conditions. The components from the first step in IM are then integrated to form a logic model of the problem, which is used to identify program objectives and thereby direct the focus of the health intervention [20]. The logic model of the problem is used to guide the next steps in IM, covering program design, program production, implementation planning, and evaluation planning [20,21].

### 1.3. The Current Study

Canteen, the national Australian organisation supporting young people (12–25 years) impacted by cancer, aimed to use IM to develop an Education and Career Support (ECS) Service for AYAs diagnosed with cancer. This paper reports on the completion of the first step in IM, as the initial step in the ECS Service’s development. The first step in IM was completed in three phases, which aimed to collect data from the literature and the target population and integrate these data into the logic model of the problem (Figure 1). Phase I was an integrative review to better understand how cancer and its treatment impacts AYAs’ capacity to engage in education and employment, as well as identify the factors contributing to, and resulting from, these impacts. Phase II utilised a cross-sectional survey to assess AYAs’ participation in education and employment, goals, career development processes, and factors associated with participation. Phase III involved collaboration with a planning group to expand upon and integrate the data collected in Phase I and Phase II into the logic model of the problem. The completed logic model of the problem will guide future program development through its description of the influences of health and demographic characteristics, behaviours, environmental conditions, and individual determinants on AYAs’ participation in education and employment. 

## 2. Phase I—Integrative Review

### 2.1. Materials and Methods

An integrative review methodology was chosen to undertake Phase I. As opposed to systematic review methodology which only allows the inclusion of experimental studies [26], integrative reviews are characteristically broad in nature, providing a critical analysis of empirical and theoretical literature and inclusive of diverse scientific methodologies [26,27]. The integrative review approach enabled us to maximise the breadth and depth of information captured about the impact of cancer and its treatment on AYAs’ capacity to engage in education or employment. Conducting an integrative review was therefore particularly suited to populating the logic model of the problem, including identifying contributing behaviours, environmental conditions, and their determinants, as well as the impact of changed capacity to participate in education and employment on AYAs’ quality of life.

#### 2.1.1. Search Strategy

Medline, PSYCInfo, and CINAHL databases were searched for peer reviewed literature using free text and MeSH terms combined with Boolean operators. Consistent with the Population Intervention Comparison Outcome (PICO) mnemonic and alternatives, these terms were focused on the population (AYAs), exposure (cancer), and outcome (education, employment, and career). For the purposes of this paper, education included any form of formal study or vocational training, employment included any form of paid work, and career referred to a chosen field of work experience that is to be achieved and cultivated over the long term. Grey literature was explored using a Google search for “AYA cancer” relating to education, employment, career, and vocation. Finally, the reference lists of included documents were manually searched for additional literature. 

#### 2.1.2. Study Selection

All literature published between 2011 and 2021 and in English were included. The time range of the previous ten years was chosen to promote currency of the results. Literature were excluded if: (i) the sample or population of interest had a mean age of 14 or younger or 40 or older at time of diagnosis, minority inclusion of AYAs aged 15 to 39 years at diagnosis in literature reviews or meta-analyses, or age at diagnosis not reported; (ii) the sample or population included individuals with diagnoses other than cancer; (iii) it reported on unpaid employment; or (iv) it reported on intervention for cancer or cancer prevalence as opposed to impact and the context of a cancer diagnosis and its treatment. 

All searches were transferred to the systematic review software Covidence [28]. The titles and abstracts of studies were screened for eligibility and duplicates removed.

#### 2.1.3. Data Extraction and Synthesis

A template within Covidence was used to extract information for each study, including the study sample or population, methodology, and results. All quantitative and qualitative results relating to the impact and context of a cancer diagnosis and its treatment on education and employment for AYAs were considered. Results from the Covidence data extraction were exported to an Excel spreadsheet and a narrative synthesis conducted. The key findings from each paper were summarised and organized according to the education, employment, or career outcome or experience (e.g., employment rates, losses to career or education aspirations) and factors found to be associated with these (e.g., sociodemographics, mental health, and social support). 

### 2.2. Results

A total of 2710 documents were screened and 2680 were excluded from the review, resulting in a total of 30 documents (Figure 2). Of the 30 studies included, 15 were quantitative studies (including 1 dissertation) [17,29,30,31,32,33,34,35,36,37,38,39,40,41,42], 11 were qualitative studies [43,44,45,46,47,48,49,50,51,52,53], 3 were literature reviews [1,9,54], and 1 was an information resource [55].

#### 2.2.1. Impact on Capacity to Participate in Education or Employment

A range of outcomes pointed to reduced AYA capacity to participate in education or employment because of cancer and its treatment. When comparing education or employment outcomes for AYAs with a history of cancer versus those without a history of cancer, there was mixed evidence on employment rates or level of educational attainment (below vs. on par or above [1,33,38]). However, there was greater reported prevalence of disability among AYAs with a history of cancer compared to AYAs without a history of cancer [9] and suggestions that survivors may have worse levels of cognitive and occupational function, including worse levels of work output [38]. This aligns with more consistently reported evidence about suboptimal employment outcomes, including more often working part-time or being older when starting first occupation, compared to AYAs without a cancer history [1,33]. 

#### 2.2.2. Factors Associated with Participation in Education or Employment

##### Individual

Individual factors leading to and the impact to quality of life resulting from reductions to AYA participation in education or employment were pervasive and complex. Financial distress was described as having long-term implications for AYA survivors’ physical and mental health [47,54]. Higher rates of mental disorders and higher levels of psychological symptoms (e.g., fatigue, anxiety, and depression) were associated with higher rates of unemployment [32]. Conversely, persistent and long-term absence from work or study was associated with poorer psychological outcomes [31]. Fears in relation to work or study were common, such as feeling “left behind” compared to peers or a sense of “missing out” [44,48] and being unable to keep up with expectations to perform well in comparison with peers [43,52]. Such fears or concerns were identified as contributing to a sense of urgency to resume school and work [44] and to “catch up” [48], even if perhaps they may regret and perceive this action as premature afterwards [43]. Losses around future aspirations, feelings of hopelessness, and uncertainty about the future were found to negatively impact motivation and ability to make career decisions [43,45,48,53]. Similarly, losses to normality [48] and feeling different to peers and pre-cancer selves [9,43,48,49,54] could be significantly challenging to AYAs’ sense of identity and in some cases lead to challenges with social re-integration and attendance [9,43,49]. Providing support to improve AYAs’ knowledge, skills, self-efficacy, and emotional wellbeing appeared critical for improving AYAs’ participation in their education and employment. 

##### Environment

Environmental factors that impact AYAs’ participation in education or employment were found at the interpersonal, organizational, community, and societal levels. These environmental factors included unsuitable demands of the job or study tasks [30,34], obstructive administrative and coordinating structures and processes within education institutions and workplaces [44,46,49], the level and appropriateness of support received from the AYAs’ local or proximal environment (e.g., family, peers and colleagues, treatment team, education or work providers [1,48,49,50]), and more distal constraining or discriminating influences at the systemic level (e.g., access to ECS services [44,46,49]). Mixed reactions to implementing work or study accommodations were reported by survivors, with some describing them as helpful and others as restrictive and counter to the desire to be treated normally and similarly to employees without a history of cancer [52]. Occupational health professionals reported conflict between supporting survivors’ work ability using various accommodations and closing the gate to employment should survivors be unable to perform essential job functions and accommodation efforts be ineffective or unfeasible [51]. There were consistent indications of the need to improve the knowledge and skills of individuals within AYAs’ environment, with the view to better understanding how to support and advocate for AYAs’ education and career needs. There were also indications of the need to improve AYAs’ access to financial, legal, psychological, and education and career services and improve continuity and coordination of care in their community. 

##### Sociodemographic and Health

A range of sociodemographic and health characteristics were associated with reduced capacity of AYAs to participate in education or employment. These included female gender [1,32,33,34], younger age [33,36,37], general health [32,44], and mental health [32]. Given the heterogeneity of diagnoses [1,35,39,41] and treatment types [1,34] examined across studies, we cannot comment conclusively on which diagnoses and treatment types caused consistent challenges to AYAs’ participation in education or employment relative to others. There was consistent evidence for more generic characteristics, such as treatment intensity [32,33,39], with higher levels of treatment intensity being associated with AYAs’ perceptions of worse work ability [32], report of greater work struggles [32], and reduced likelihood of returning to full-time work or study [39] or graduating from high school or university [33]. AYAs frequently attributed the presence and level of severity of cancer late effects as a reason for reduced capacity to work or study [32,36,48,49,53], including their ability to attend and engage with school [46,48] and to re-enter work or study successfully and sustainably [44].

## 3. Phase II—Quantitative Survey

### 3.1. Materials and Methods

#### 3.1.1. Study Design

Phase II utilised a cross-sectional survey to assess current education and employment participation, goals, career development, and perceived barriers in a community-based sample of AYA-aged people who had been diagnosed with cancer in childhood, adolescence, or young adulthood. 

#### 3.1.2. Procedure

Phase II received ethics approval from the University of Sydney Human Research Ethics Committee (Protocol Number 2021/648). Eligible participants were AYAs who had received a personal diagnosis of cancer and were aged 15–25 years at the time of accessing support from Canteen. There were no exclusion criteria based on cancer variables (e.g., age at diagnosis, cancer type, treatment type, or time since diagnosis or treatment). An invitation to participate in the study was sent to 406 eligible participants via email or SMS. This invitation contained a link to the online information sheet, consent form, and survey. Informed consent was obtained prior to participants accessing the survey. 

#### 3.1.3. Measures

##### Demographic Measures

Participant demographics were collected at the start of the survey and included information on participant age, postcode, gender identity, Aboriginal or Torres Strait Islander identity, languages spoken at home, highest level of completed education, and cancer experience, including diagnosis type, year, and current treatment status. Postcode data were used to calculate participants’ remoteness area in Australia using the Australian Statistical Geography Standard correspondence for 2017 postcode to 2016 Remoteness Area. This classifies participants’ residential area into five remoteness classes based on their access to services, ranging from Major City to Very Remote [56].

##### Education and Employment Participation

Participation in education was assessed through multiple-choice questions on participants’ current enrolment status and enrolment location. Response options for enrolment status were not enrolled, not looking for opportunities; not enrolled, looking for opportunities; enrolled part-time; and enrolled full-time. Response options for enrolment location were school, TAFE, vocational training, university, or other, with a free text option to respond. 

Participation in employment was assessed through a multiple-choice question on the participants’ current paid employment. Response options were not employed, not looking for work; not employed, looking for part-time/casual work, or looking for full-time work; working in unpaid employment; employed casually, part-time, or full-time. 

Participants’ education and employment status was combined to calculate their engagement in education and employment as per the methodology for the Survey of Education and Work [57]. Participants were classified as ‘Not engaged’ if they were not participating in education or paid employment; as ‘Partially engaged’ if they were participating in either part-time education or employment; and as ‘Fully engaged’ if they were participating in full-time education, full-time employment, or both part-time education and employment. Participants status as being in or out of the labour force and the employment, unemployment, and underemployment rates were calculated using definitions and data from the Australian Bureau of Statistics [58]. Participants were also asked how many days they were unable to carry out their usual education or employment activities in the past 2 weeks due to their cancer experience, with response options of 0 days, 1–3 days, 4–6 days, 7–9 days, and most or all days.

##### Education and Employment Goals

The participants’ goals for education and employment were assessed separately. For each, the participants’ current goals were assessed using a multiple-choice question with a free text option. Response options included no goals; increase hours; stay in/complete current; start new; find career-related alternative; or, for employment only, increase pay. Participants reported their confidence in achieving these goals on an 11-point Likert scale from 0 = not at all confident to 10 = extremely confident. 

The participants reported whether they had changed their goals because of their cancer diagnosis or treatment on a 4-point scale with options for no change and small, moderate, or large change. A binary variable was created for education and employment by retaining the no change option and combining the small, medium, and large change options. The participants who reported any change in their goals were asked to provide a free text response describing the change. This text was reviewed and coded by author KC as a goal reduction (e.g., moving from full-time to part-time work) or an expansion (e.g., adding values-aligned subjects to study course). Responses where the direction of change was not obvious were not coded.

##### Career Development Processes

Confidence in career development processes was assessed using items from the Children’s Brain Tumor Foundation Career Assessment [59]. Four items assess career/education awareness, including participants’ knowledge and confidence in making effective education and career decisions and close people’s support with and beliefs in the importance of education and career. These four items were combined to create a career awareness subscale; in this study, Cronbach’s α was 0.832, indicating high internal consistency. One item assessed confidence in career acquisition and one item assessed confidence in career maintenance. All items were measured on a 5-point Likert scale from 1 = strongly disagree to 5 = strongly agree. 

##### Barriers to Education and Employment

Factors that could impact AYAs’ education and employment were measured using an adapted version of the Perceived Barriers Scale (PBS) [60]. The adapted measure contained 13 items, with participants asked to rate how much each item is a barrier to achieving their education and/or employment goals on an 11-point Likert scale, with anchors of 0 = not a barrier, 5 = somewhat of a barrier, 10 = major barrier. A binary variable was created for each item by treating scores below the midpoint as ‘not a barrier’ and scores at or above the midpoint as a ‘barrier.’ The original measure has two subscales, assessing internal barriers (e.g., mental, physical, social, and cognitive functioning) and external barriers (e.g., attitudes and support from other people) to goals. Item responses are summed to calculate subscale scores, with higher scores indicating higher perceived barriers. In this study, the subscales demonstrated a high level of internal consistency, with a Cronbach’s α of 0.85 for the internal barriers subscale and 0.81 for the external barriers subscale.

##### Impact of COVID-19

The impact of the COVID-19 pandemic on participants’ education, employment, financial situation, and confidence in goal achievement was assessed using a 5-point Likert scale, ranging from 1 = a lot harder/worse to 5 = a lot easier/better.

#### 3.1.4. Data Analysis

Descriptive statistics were used to summarise categorical and ordinal variables. Spearman Rank Order Correlations were used to assess associations between engagement in education and employment; education goal confidence; employment goal confidence; the three career development processes of career awareness, acquisition, and maintenance; and internal barriers and external barriers to goal achievement. The Benjamini–Hochberg procedure was used to limit the false discovery rate to 0.05 for the 28 correlations between these variables [61]. After applying the Benjamini–Hochberg procedure, the index *p*-value for significance was determined to be 0.038. Spearman Rank Order Correlations with the Benjamini–Hochberg procedure applied were also used to explore associations between engagement in education and employment and the individual items on the career awareness, internal barriers, and external barriers subscales. The index *p*-value for significance for these 17 correlations was determined to be 0.022.

### 3.2. Results

#### 3.2.1. Demographics

Of the 406 eligible participants, 82 (20%) completed the survey and were included in the study. The mean age of the participants at the time of the study was 20.1 years (*SD* = 3.04) (Table 1). At time of diagnosis, the mean age was 16.3 years (*SD* = 5.39) and the mean length of time since diagnosis was 4.31 years (range 0–20 years). Most participants (*n* = 67, 83%) had completed treatment at the time of the survey. There were a mix of cancer diagnoses, with the three most reported diagnoses being lymphomas (Hodgkin or non-Hodgkin; *n* = 22, 27%); leukaemia (*n* = 19, 23%), or brain or central nervous system cancers (*n* = 14, 17%). This represents a higher proportion of lymphoma or leukaemia diagnoses compared with the general population of AYAs diagnosed with cancer (20% and 7%, respectively) [62].

#### 3.2.2. Participation in Education and Employment

Two-thirds of participants were enrolled in education either full-time (*n* = 35) or part-time (*n* = 19). Of those in the labour force (*n* = 69, 84% of sample), nearly three-quarters were working in paid employment at the time of the survey (*n* = 51, 74% of labour force). Most were employed part-time (*n* = 14, 20%) or casually (*n* = 26, 38%) rather than full-time (*n* = 11, 16%). Combining education and employment status, two-thirds of participants (*n* = 54, 66%) were classified as fully engaged in education or employment, 22% (*n* = 18) as partially engaged, and 12% (*n* = 10) as not engaged. 

Around half of participants (*n* = 42, 51%) reported that they had been unable to carry out most of their usual education or employment activities for at least one day in the last two weeks due to their cancer experience. This included 10 participants (12%) who reported their cancer experience had disrupted their usual activities on most or all days in the last two weeks. 

#### 3.2.3. Education and Employment Goals

Sixty-nine participants (84%) reported having a current goal for their education. For these participants, the most frequently reported goal was to complete their current education (*n* = 53, 77%). Other reported goals were to find career-related education (*n* = 18; 25%); start participating in education (*n* = 10; 14%); or increase hours in education (*n* = 9; 13%). Nearly half reported having high confidence in achieving their education goals (*n* = 31, 46%), with 42% reporting medium confidence (*n* = 28) and 10% reporting low confidence (*n* = 7). Around two-thirds reported they had made a change to their education goals because of their cancer diagnosis or treatment (*n* = 53, 65%). Of these, 86% (*n* = 38) described reducing their goal in some way; for example, reducing their study load or hours, or switching to a less intensive field of study. Fourteen percent (*n* = 6) described expanding their goals in some way; for example, changing courses to study a healthcare profession.

Seventy participants (85%) reported having a current goal for their employment. For these participants, the most frequently reported goal was to find different employment more closely related to their career (*n* = 26, 32%). Other reported goals were to stay in their current employment (*n* = 33%); start participating in employment (*n* = 20; 29%); increase employment hours (*n* = 16; 23%); or increase pay (*n =* 6; 9%). Around one-third reported having high confidence in achieving their employment goals (*n* = 21, 31%), with 52% reporting medium confidence (*n* = 35) and 16% reporting low confidence (*n* = 11). Just over half reported they had made a change to their employment goals because of their cancer diagnosis or treatment (*n* = 47, 57%). Of these, 88% described reducing their goal in some way; for example, reducing work hours, or switching to a less physical type of work. Twelve percent described expanding their goals; for example, pursuing a career aligned with their values. Half the participants had changed both their education and employment goal (*n* = 41, 50%), while 22% (*n* = 18) had changed only one goal and 28% (*n* = 28) had changed neither goal.

#### 3.2.4. Career Development Processes

At least 60% of participants agreed or strongly agreed with all statements about their career/education awareness, acquisition, and maintenance abilities (range 60–92%), with over 90% agreeing or strongly agreeing that people close to them believe in the importance of their education and career (Figure 3). The two items with the lowest proportion agreement were knowledge-based (both 60% agreement), i.e., knowledge about information-seeking and knowledge about how to find a relevant job. 

#### 3.2.5. Barriers to Education and Employment

The mean score on the internal factors subscale of the adapted PBS was 27.25 (*SD* = 15.186; range 0–58) and on the external factors subscale was 12.85 (*SD* = 12.698; range 0–46). Over 60% of participants reported the individual factors of cancer symptoms, physical health, and mental health as a barrier to education and career goal achievement (Figure 4). External factors were reported as a barrier by fewer than a third of participants. 

#### 3.2.6. Impact of COVID-19

Participants reported negative impacts of the COVID-19 pandemic on their education (*n* = 53, 71%); employment (*n* = 41, 49%); optimism about the future (*n* = 46, 58%); confidence in achieving future goals (*n* = 38, 48%); and financial situation (*n* = 32, 44%).

#### 3.2.7. Correlational Analysis

Factors that were significantly correlated with greater participation in education and employment were higher education and employment goal confidence (*rho* = 0.281, *p* = 0.022; *rho* = 0.391, *p* = 0.001, respectively), greater career awareness (*rho* = 0.405, *p* < 0.001), and fewer internal and external barriers to goal achievement (*rho* = −0.263, *p* = 0.019; *rho* = −0.288, *p* = 0.009, respectively). There were also a range of small to medium correlations between and within the variables of goal confidence, career development, and barriers; see Table 2.

Exploratory correlations between education and employment participation and individual items on the career development processes and barriers measures found greater participation was correlated with higher agreement with the three career awareness items that close people provide support with education and career decisions (*rho* = 0.372, *p* = 0.001); close people believe in the importance of education and career (*rho* = 0.311, *p* = 0.007); and self-confidence in the ability to make education and career decisions (*rho* = 0.305, *p* = 0.007). Greater participation was also correlated with lower perceived internal barriers from skills to capably participate in study or work (*rho* = −0.289, *p* = 0.009); physical health and functioning (*rho* = −0.284, *p* = 0.01); and cancer symptoms or side effects (*rho* = −0.285, *p* = 0.01, as well as lower perceived external barriers from unsupportive employers (*rho* = −0.296, *p* = 0.007); and access to education providers (*rho* = −0.252, *p* = 0.022). 

## 4. Phase III—Planning Group and Development of the Logic Model of the Problem

### 4.1. Materials and Methods

Phase III involved recruiting and convening a planning group of key stakeholders to provide input and guidance on the development of the ECS Service. The planning group leveraged their knowledge and experience to expand upon and integrate Phase I and Phase II data into the logic model of the problem. Phase III received ethics approval from the University of Sydney Human Research Ethics Committee (Protocol Number: 2021/648].

#### 4.1.1. Planning Group Procedure

We used purposeful sampling to ensure participants in the planning group were representative of stakeholders at the individual, interpersonal, organisational, community, and societal levels [19]. The choice of stakeholder representation was in recognition that the education and career needs of AYAs diagnosed with cancer would be more relevant if informed by AYAs and representatives of those in their environment who have an influence on their education and career behaviours. 

Individuals invited to participate in the planning group were AYAs and parents accessing Canteen services, staff employed by Canteen, and external professionals. AYAs were aged 15 to 25 years at the time of the study and had a diagnosis of cancer during their lifetime. AYAs were invited through direct approach from Canteen staff or by indicating their interest at the end of the Phase II survey. The parents of AYAs were invited via an advertisement in a Canteen online parent support platform. Canteen staff and external professionals were directly approached via an email from the researchers. Canteen staff included those across general, clinical, community education, and education and career services. External professionals included those across clinical, research, advocacy, and education and career services or organisations. Interested individuals were offered the opportunity to receive additional information about participation before providing written informed consent. 

A total of 20 stakeholders agreed to participate in the planning group. These stakeholders included five AYAs; one parent of an AYA who had undergone treatment for cancer; six Canteen staff; and eight external professionals. 

The planning group met twice in the lead up to developing the logic model of the problem. Meetings were conducted via videoconference and utilised an online whiteboard platform, to present information and facilitate brainstorming activities. Each meeting was prefaced by a clearly stated agenda, and the content and processes of each meeting were summarised in meeting minutes. In the first meeting, the outline and scope of the project was discussed, including IM methodology. During the second meeting, the research team presented results from Phase I and Phase II. The planning group discussed the presentation findings and then worked together to develop the logic model of the problem. In addition to these meetings, the planning group routinely contributed to discussions via post-meeting feedback forms and individual conversations with the researchers.

#### 4.1.2. Development of the Logic Model of the Problem

The research team prepared a template of the logic model of the problem to populate within the second planning group meeting. The logic model was developed from right to left, beginning with descriptions of quality of life and health problem [20]. The research team pre-populated the definition of the health problem as “no or part-time participation in education or employment” based on the Phase I and Phase II results. Namely, consistent findings in the literature about the prevalence and impact on AYAs’ quality of life from impaired attendance or engagement with education or employment [1,5], as well as the finding that one-third of Phase II participants were not, or only partially, engaged in education or employment The research team then inputted the quality of life issues and health and demographic factors associated with “no or part-time participation in education or employment” based on the Phase I and Phase II results.

During the second meeting, the research team demonstrated what a logic model of the problem is and how the planning group and research team would develop a model that explains “no or part-time participation of AYAs in education or employment”. The research team then presented the Phase I and II findings and subsequent rationale for the definition of the health problem and the selection of associated quality of life issues and health and demographic factors. The planning group were first invited to comment on the appropriateness and relevance of this pre-populated content. Moving to the left in the model, the planning group were then guided in brainstorming behaviours, environmental conditions, and their determinants that may contribute to the health problem. 

The researchers collated the feedback of the planning group from the meetings, feedback forms, and individual conversations. The researchers then refined and finalised the model through further synthesis with Phase I and Phase II findings and collaborative consensus. 

### 4.2. Results

#### 4.2.1. Planning Group Feedback

Upon consideration of the Phase I and Phase II findings, the planning group indicated broad agreement with the choice of the health problem as “no or part-time participation in education or employment”, the extent and scope of the identified quality of life issues, and selection of health and demographic factors that influence participation. The planning group emphasised the impact of a life-changing diagnosis on AYAs’ sense of purpose and engagement in education or employment, such that “no or part-time participation” primarily affects those whose participation is misaligned with their values, goals, and preferences. 

From subsequent brainstorming activities, the planning group provided unique and up-to-date insights about behavioural and environmental factors and their determinants that complemented and expanded upon what was identified from the literature review and survey. Insights included the lack of supportive policies in Australia for AYAs impacted by cancer around accessing long-term support for survivorship needs, including education and career. The planning group also highlighted a lack of understanding and support of survivors’ unique needs among education, training, and work providers. There were also insights about the current gaps experienced by AYAs and family in detecting and supporting education and career needs across the survivorship trajectory, and the reliance on their help-seeking behaviours and self-advocacy to have such needs met. The planning group feedback highlighted how specialised the knowledge was of identifying, accessing, and navigating support for education, employment, and career needs.

#### 4.2.2. Logic Model of the Problem

Figure 5 displays the logic model of the problem, including the influence of health and demographic characteristics, behaviours, environmental conditions, and individual determinants on AYAs’ participation in education or employment. The definition of the health problem was expanded to “no or part-time participation in personally meaningful education or employment”, in acknowledgement of the importance of AYAs engaging in education or employment that is aligned with their values, goals, and preferences.

At the individual level, behaviours identified as contributing to the health problem included low attendance or engagement with education or employment; fragmented approach to education or employment tasks and goals; low quality relationships or engagement with peers, colleagues, employers, or educators; poor self-management of the impacts of cancer; and limited help-seeking. The determinants behind such behaviours were focused on the levels of AYAs’ knowledge, skills, self-efficacy, and emotional wellbeing surrounding their cancer and education and career. 

At the environmental level, the key influencers on AYAs’ behaviour were recognised as family, peers and colleagues, employers and education providers, healthcare providers, and policy makers, legislators, and advocacy groups. The environmental conditions identified as impacting AYAs’ participation in education or employment centred on provision of no or inappropriate support to meet AYAs’ education and career needs; limited access to supportive services; low availability of continuous and coordinated cancer and survivorship care; and various systemic structures and policies within the education, employment, and health systems. The determinants contributing to such conditions among the key influencers included their levels of knowledge, skills, and self-efficacy about, and attitudes towards, supporting AYAs’ education and career needs.

## 5. Discussion

### 5.1. General Results

The logic model of the problem presents factors that reduce AYAs’ participation in meaningful education, employment, and careers and thus can be used a framework to guide the development of tailored education and career intervention. The need for evidence-based education and career support for AYAs with a cancer history is supported by confirmation that “no or part-time participation in personally meaningful education or employment” is a health problem for AYAs, with flow-on effects for their quality of life. The behaviours, environmental conditions, and their determinants that contribute to the health problem indicate specific needs that can be targeted through such an intervention.

### 5.2. The Health Problem and Impact on Quality of Life

The health problem selected by the planning group for the logic model of the problem was “no or part-time participation in personally meaningful education and employment” in AYAs aged 15–25 years after a cancer diagnosis. The literature reviewed in Phase I described that AYAs with a cancer history experience significant challenges with attendance at school or study and with engaging in employment following their diagnosis. The results from the Phase II survey indicated that the health problem was present for AYA clients of a community-based organisation. A lower proportion of AYAs reported being fully engaged in education or employment compared to the general Australian population of young people aged 15–24 years (66% and 81%, respectively), while a higher proportion reported being partially engaged (22% and 10%, respectively) [63]. 

AYAs diagnosed with cancer and their same-age peers reported similar rates of high school completion (93% and 90% of people aged 20–24 years, respectively, had completed Year 12 or beyond) [63] and current enrolment in education or training (66% and 65%, respectively) [63]. However, many participants described reductions in the pace of completion (e.g., moving from full-time to part-time study after cancer) or level of enrolment (e.g., switching from a double to a single degree), indicating impacts of cancer on how AYAs are engaged in education. AYAs diagnosed with cancer also reported a higher rate of unemployment compared to their peers (26% and 11%, respectively), with a concomitant lower rate of employment (74% and 89%, respectively), particularly lower full-time employment (16% and 40%, respectively) and higher part-time employment (58% and 49%, respectively) [64]. Further, 29% of employed AYAs were regarded as underemployed, due to seeking more hours within their current employment, compared to a general youth underemployment rate of 18% [65]. This lack of, or limited, engagement in education and employment presents a health problem for AYAs due to its association with poor well-being and lower quality of life [1]. 

The planning group in Phase III specifically highlighted reduced participation in ‘personally meaningful’ education and employment as a health problem for AYAs. Previous studies have found that 50% of AYAs have reported modifying or adjusting their education and vocation goals after cancer [53] and almost 50% have reported not being “back on track” with their pre-diagnosis education and employment plans up to 24 months post-diagnosis [16]. These findings are consistent with the Phase II survey results, with half the survey respondents reporting changing both their education and employment goals after their diagnosis and a further 22% changing either their education or their employment goal. The majority of AYAs described needing to reduce their education and employment goals to fit their physical, cognitive, or emotional functioning after cancer. These functional constraints can cause a shift away from personally meaningful goals held prior to a cancer diagnosis, as described by many AYAs who had to change study course, reduce work hours, or engage in less demanding work after their cancer diagnosis. These shifts can create goal conflict between what is possible and what is valued. Goal conflict is associated with reductions in AYAs’ well-being and quality of life [9,53]. Conversely, well-being and quality of life can be supported through expanding goals to fit new values and motivations [9,53]. While full-time engagement in education and employment has been considered ideal due to its association with higher quality of life, enabling AYAs to participate in line with values-based goals may have a similar impact on quality-of-life outcomes. 

### 5.3. Behavioural Determinants of the Health Problem

The logic model of the problem presents several behaviours or actions by the individual AYA that may contribute to limited participation in education and employment. The behavioural determinants identified in the logic model of the problem represent specific support needs for AYAs, spanning their knowledge, beliefs, skills, self-efficacy, or emotional functioning. Targeting these behavioural determinants through an education and career intervention could support AYAs to engage in productive behaviours, thereby increasing participation in education and employment and ultimately improving quality of life. 

Emotional well-being was identified as a key determinant of several behaviours in AYAs that limit participation in meaningful education and employment, including social avoidance of peers or colleagues or premature return to work or study. For example, AYAs experiencing high levels of psychological distress or anxiety after cancer might find it challenging to engage with education or employment, whereas those with fears of missing out or negative evaluation may prematurely return to study or work to avoid missing out on further opportunities and social connection [1]. Thus, the logic model of the problem suggests that addressing unmet emotional support needs could lead to reduced avoidance, greater social connection, and increased participation in education and employment. At the same time, it is widely accepted that participating in education and employment can contribute to improved emotional well-being for AYAs [9], highlighting the bidirectional relationship between emotional well-being and engagement in education and employment. Improving emotional well-being may increase AYAs’ ability to engage, leading to further improvements in emotional well-being. AYAs may benefit from services or resources that help them to manage difficult emotions, or to enhance positive meaning-making, such as post-traumatic growth, positive goal re-evaluation, or identity formation [46,52]. AYAs’ ability to manage social situations with peers and colleagues may also be enhanced through the provision of information resources about disclosing a cancer history or through social support groups that normalise the emotional impact of cancer. 

A contributing factor to AYAs’ participation in education and employment is their level of career awareness, i.e., their awareness of their skills and values, confidence in making career decisions that align these skills and values with their environment, and confidence in being supported by others in these decisions [8]. Career awareness interventions aim to increase a young person’s understanding of their personal identity, skills, and values. This is likely to be important for AYAs, as cancer can disrupt identity formation through missed opportunities, or enhance personal identity, meaning, or values discovery through post-traumatic growth [47,53]. AYAs’ ability to make career decisions is also influenced by their knowledge of available career pathways and ability to select careers and level of engagement in education and employment that aligns with their personal identity and values. Improving AYAs’ ability to make ability- and values-aligned career decisions could help to increase AYAs’ confidence in goal achievement, which was a need established in the Phase II sample. Specifically, only one-third to one-half of participants in Phase II reported high confidence in their ability to achieve their education or employment goals. As difficulty achieving goals is associated with poor emotional well-being and lower quality of life, helping AYAs to set meaningful goals and build self-efficacy in overcoming goal barriers may help to improve their quality of life [53]. AYAs could be supported with values discovery, identity formation, and aligning identity and goals with potential career pathways through coordinated care between ECS and mental health services. By ensuring AYAs can make informed choices about education or employment that are aligned with their interests and abilities, it is hoped that they will be able to participate in education and employment more fully and for the longer term. 

Both the Phase I literature review and the Phase II survey results highlighted the association between limited participation in education and employment and health or demographic factors, for example, associations with treatment intensity [32,33,39], presence and severity of late effects [32,36,48,49,53], poor general health [32,44], or female gender [1,32,33,36]. These factors should be considered when determining the likelihood of AYAs needing support or prioritising more vulnerable groups to receive support. The provision of information resources on cancer symptoms, side effects, and late effects could enhance AYAs’ knowledge, skills, and self-efficacy in self-management of their physical health and well-being to maintain participation in education or employment [59]. 

### 5.4. Environmental Determinants of Outcomes

The logic model of the problem presents several environmental conditions across the interpersonal, organisation, community, and societal levels that may contribute to limited participation by AYAs in education and employment. The environmental determinants indicate specific support needs for knowledge, skills, motivations, and self-efficacy of individuals within those levels, including peers and colleagues, educators, employers, healthcare providers, and policy makers. Due to the bidirectional relationship between AYAs’ behaviours and their environmental conditions, intervening at the environmental level can directly and indirectly serve to improve AYAs’ participation in education and employment. 

The lack of access to education, employment, and career support services and resources was identified as a barrier to AYAs’ participation in education and employment. Developing policies that dedicate resources to increased service provision could markedly improve AYAs’ participation in meaningful education and employment. This may be particularly important for AYAs who have general access issues, for example, those who are socioeconomically disadvantaged or living in rural and remote areas [66]. 

In addition to access to education, employment, and career support services, it is equally important to focus on the quality-of-service provision. This quality is driven by the knowledge, skills, motivations, and self-efficacy of support providers. In the absence of dedicated or specialist services, the unmet need for education and employment support after cancer is often addressed through informal support sources [67], particularly family [16]. While the population of AYAs with a cancer history is growing, it is still a fortunately rare occurrence, but this limits the knowledge of its impacts within the general population. Targeting support provider knowledge of AYA cancer impacts and ways to support AYAs is an important first step in improving the environments in which AYAs live. For example, provision of information resources to employers, education, or healthcare providers may increase the understanding of the need to provide support to AYAs engaging in education or employment after cancer and increase the motivation to do so [46]. An equally important step is to provide support that improves providers skills and their confidence to use those skills (e.g., in responding to a cancer diagnosis from a young person or setting realistic accommodations in education and employment settings). Improving the knowledge, skills, motivations, and self-efficacy of providers may also improve coordination and continuity of care, as providers share their knowledge and experience across AYA networks and communities. 

### 5.5. Strengths and Limitations

This study is strengthened by using IM as the process for identifying the education and employment support needs of AYAs. The IM process requires the detailed elucidation of both individual and environmental conditions and their determinants. There is an acknowledged gap in the literature on the predictors of AYAs’ engagement in education and employment after a cancer diagnosis [1]. Two known existing frameworks—the IWWM [8] and adapted CMOP-E [9]—describe individual and environmental factors and determinants that contribute to education and employment outcomes; however, the logic model of the problem developed as part of this study provides additional depth and specificity of detail to what is provided in these frameworks. This detail is a direct result of the comprehensive protocol of the first step in IM, with data triangulated across a literature review, survey, and participatory planning group. As a result, the detail in the logic model of the problem, especially with respect to determinants, allows the identification of specific needs that could be targeted through an education and career intervention, as well as inform service planning and outcome measurement. The logic model was also developed with the needs of a broader group of AYAs diagnosed with cancer in mind (i.e., AYAs who have or have had cancer at any age) than what is represented in the IWWM (young adult survivors of childhood cancer) and adapted CMOP-E (AYAs with current cancer), and with the intention of addressing these needs within a community-based organisation. Finally, another strength of using IM was that the content came from a participatory process involving key stakeholders who held a variety of perspectives, spanning young people and parents with lived experience, healthcare providers, employers, and education and career advisors. Gaining this variety of perspectives has leant confidence in the validity and acceptability of the model. In turn, we have greater confidence in our understanding of several major challenges that AYA face in their education and employment following a cancer diagnosis, and modifiable causes of those challenges. 

There are several limitations to the study that should be acknowledged. First, the measurement of education and employment outcomes in Phase II occurred during the COVID-19 pandemic at a time of known impact to participation in wider society. The participants described a negative impact of COVID-19 on their education, employment, and confidence in achieving their future goals. However, this timing is offset by the fact that comparative statistics were also recorded during the pandemic and as close as possible to the time of survey data collection. A further limitation of the study is the relatively small sample size for Phase II, which precluded the investigation of differential impact on outcomes based on participant characteristics. However, these factors are perhaps the most well supported within the existing literature and therefore were the area least requiring new insights from either the survey or planning group. Further, the study examined bivariate associations between variables in Phase II, rather than examining a predictive model. At this phase of needs assessment, bidirectional associations between outcomes and predictive variables were of most interest; predictive models of outcomes will be incorporated into the future evaluation of the ECS Service once implemented.

### 5.6. Implications and Next Steps

This logic model has multiple implications for improving the participation in education or employment for AYAs impacted by cancer. Practitioners can use the model as a guide in assessing potential areas of need, for which they can then apply evidence-based interventions and link AYAs to necessary supports. Service planners can use the model to guide development of service goals and to compare potential AYAs’ need to current resourcing, thereby supporting prioritisation in service delivery. Policy makers and advocacy groups may leverage the model at a more systemic level, as it highlights a clear need for education, employment and career support and the impact that no or limited participation in education or employment can have on AYAs’ quality of life. This logic model is also useful as a guide for researchers who seek to examine and better understand the factors and outcomes related to education, employment, and career outcomes for AYAs with a cancer history. 

In this vein, the logic model of the problem will be used as part of a project to enhance an education and career support service at a community-based youth cancer care organisation. The upcoming stage of the project is to define the goals and objectives for the support service. We will then proceed with the next steps of IM. These steps are to select evidence-based theories of change or existing interventions to guide service development; assess fit between theories of change, intervention components and AYAs’ needs and service goals and objectives; select, design, and test service components; develop an implementation plan; and, following implementation, evaluate service effectiveness. These steps will continue to be conducted by integrating data from multiple sources, including the ongoing input of the planning group. 

## 6. Conclusions

AYAs with a cancer history are at risk of disengagement from education and employment for many reasons. The current study outlines the use of IM to develop a logic model that describes and explains the influence of health and demographic characteristics, behaviours, environmental conditions, and individual determinants on AYAs’ participation in education or employment. It is a valuable framework that can be used alongside existing similar frameworks to drive clinical, service, research, and policy improvements for AYAs’ education, employment, and career support, with the aim of improving AYAs’ broader societal participation, productivity, health, and quality of life. 

## Figures and Tables

**Figure 1 cancers-14-04590-f001:**
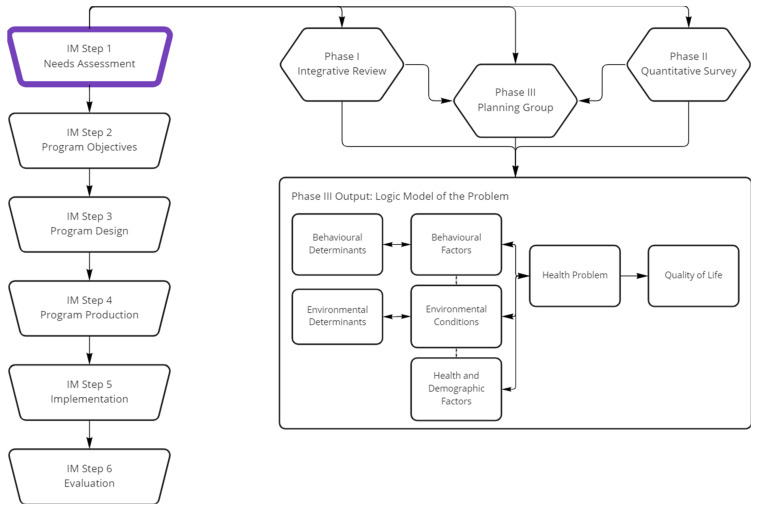
Program planning for an Education & Career Support (ECS) Service for AYAs diagnosed with cancer.

**Figure 2 cancers-14-04590-f002:**
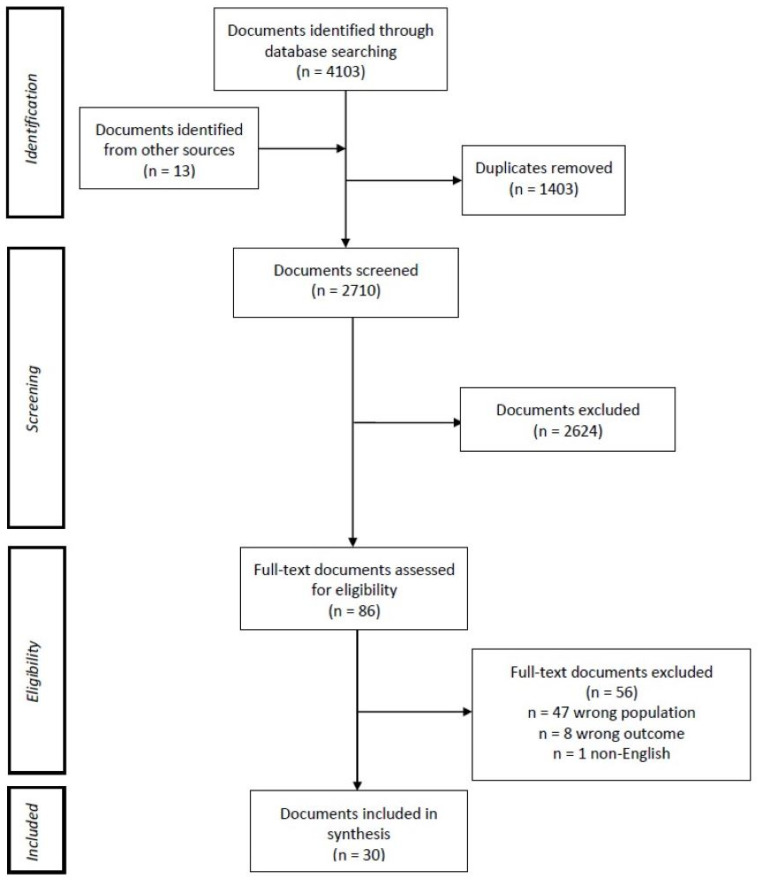
Study flow diagram.

**Figure 3 cancers-14-04590-f003:**
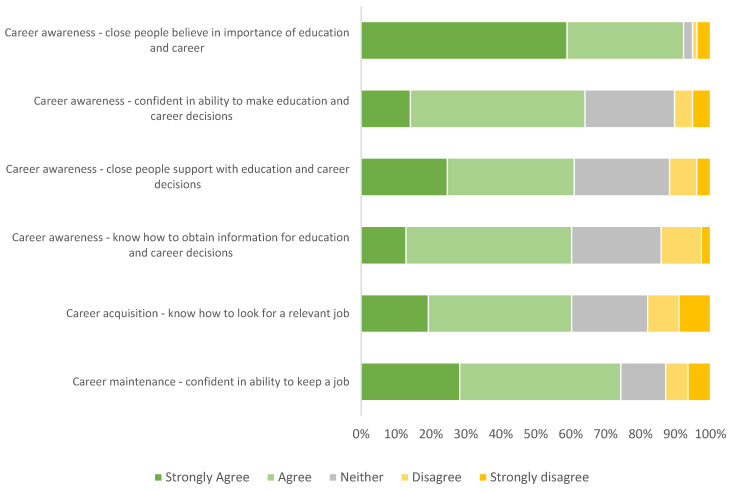
Agreement that participants have abilities in career development processes.

**Figure 4 cancers-14-04590-f004:**
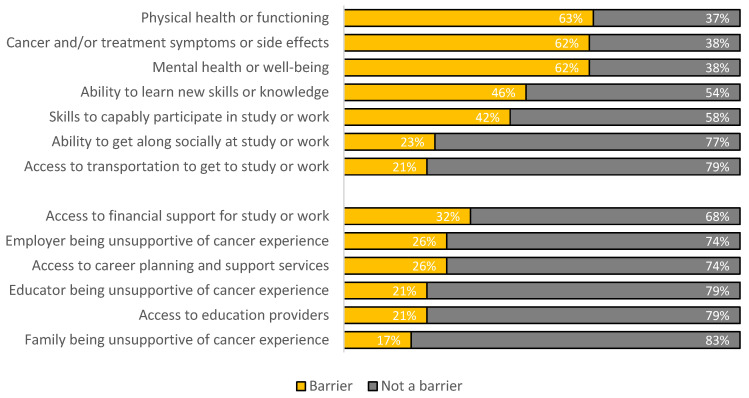
Internal and external barriers to achievement of education and employment goals.

**Figure 5 cancers-14-04590-f005:**
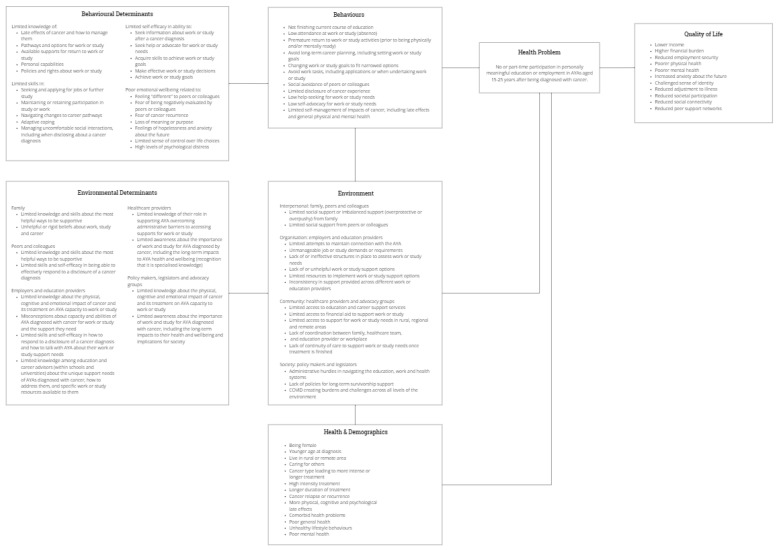
Logic model of the problem.

**Table 1 cancers-14-04590-t001:** Descriptive characteristics of the participants (*n* = 82).

Demographic Factors	
Age at survey, mean (SD)	20.1 (3.044)
Gender	
Woman, *n* (%)	62 (76%)
Man, *n* (%)	17 (21%)
Non-binary, *n* (%)	1 (1%)
Prefer not to disclose, *n* (%)	2 (2%)
Living in major city, *n* (%)	63 (77%)
LOTE speaker, *n* (%)	13 (16%)
First Nations, *n* (%)	5 (6%)
Cancer-related factors	
Age at diagnosis, mean (SD)	16.3 (5.39)
Years since diagnosis, mean (SD)	4.31 (4.582)
Current treatment status	
Active or maintenance treatment, *n* (%)	9 (11%)
Completed treatment, *n* (%)	69 (85%)
Monitoring, *n* (%)	2 (2%)
Palliative, *n* (%)	1 (1%)
Unknown, *n* (%)	1 (1%)
Education and employment factors	
Current education enrolment	
Not enrolled, not seeking opportunities, *n* (%)	16 (20%)
Not enrolled, seeking opportunities, *n* (%)	12 (14%)
Enrolled in school qualification, *n* (%)	16 (20%)
Enrolled in non-school qualification, *n* (%)	38 (46%)
Current employment	
Not in the labour force (not employed or seeking employment), *n* (%)	13 (16%)
In the labour force	69 (84%)
Unemployed	18 (26%)
Employed part-time or casually, *n* (%)	40 (58%)
Employed full-time, *n* (%)	11 (16%)
Highest education completed for post-school-age participants (20+ years; *n* = 48)	
Year 11 (or equivalent), *n* (%)	3 (6%)
Year 12 (or equivalent), *n* (%)	25 (52%)
Certificate or diploma, *n* (%)	10 (21%)
Bachelor’s degree (with or without Honours), *n* (%)	10 (21%)

**Table 2 cancers-14-04590-t002:** Correlations between measures of education and employment participation, goals, career development, and perceived barriers.

	1	2	3	4	5	6	7	8
1. Engagement in education or employment (rho, *p*)	---	0.281 ***0.022***	0.391 ***0.001***	0.405 ***<0.001***	0.145 *0.204*	0.214 0.060	−0.263 ***0.019***	−0.288 ***0.009***
2. Education goal confidence (rho, *p*)		---	0.554 <***0.001***	0.268 ***0.037***	0.237 *0.064*	0.257 *0.043*	−0.239 *0.057*	−0.199 *0.109*
3. Employment goal confidence (rho, *p*)			---	0.449 <***0.001***	0.442 <***0.001***	0.493 <***0.001***	−0.426 <***0.001***	−0.417 <***0.001***
4. Career awareness (rho, *p*)				---	0.556 <***0.001***	0.521 <***0.001***	−0.219 *0.061*	−0.332 ***0.003***
5. Career acquisition (rho, *p*)					---	0.472 <***0.001***	−0.295 ***0.010***	−0.277 ***0.014***
6. Career maintenance (rho, *p*)						---	−0.264 ***0.022***	−0.285 ***0.011***
7. Internal barriers (rho, *p*)							---	0.670 <***0.001***
8. External barriers (rho, *p*)								---

Note: Bold indicates a statistically significant *p*-value.

## Data Availability

The data presented in this study are available upon request from the corresponding author. The data are not publicly available to ensure privacy.

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
