# Peer review of "Using Intervention Mapping to Develop an Education and Career Support Service for Adolescents and Young Adults Diagnosed with Cancer: Identification of the Contextual Factors That Influence Participation in Education and Employment"

_cancers, 2022, doi:10.3390/cancers14194590_

Round 1

Reviewer 1 Report

Overall a very strong and interesting manuscript.  I think that there are 3 issues with the manuscript that I believe can be addressed. First, the description and results related to the third phase is a bit hard to follow. Specifically, I think that there needs to be more clarification on how the information identified in phase 1 & 2 were used in phase 3.  How did the participants utilize the information? how was the information presented? where the participants guided or directed to certain information?  Was the group provided a rubric or structure to analyze the information? If so, were they provided with training or orientation on how to use the rubric?  In my opinion, there is a lot of information identified in phases 1& 2 and asking a group of 20 individuals from diverse backgrounds to synthesize the information seems a heavy lift. So if the researchers did anything to help the group synthesize or analyze the information, that should be explained in a bit more detail . 

Second, this information is dense, packed with information, and a tough read due to length.  I would suggest that the authors think about ways in which the manuscript could be shortened?  Presenting information in tables may be useful.  Anything to help the reader stay focused on the information being presented would be useful.  

Finally, it might be helpful for the authors to provide some analysis of how the logic model developed is different/similar to the other models discussed at the front of the paper.  How does this model advance the prior models.  

Reviewer 2 Report

Thanks for the opportunity to review your paper entitled "Using intervention mapping to develop an education and career support service for adolescents and young adults diagnosed with cancer: Identification of contextual factors that influence participation in education and employment". Your paper described a three-stage model to inform the development of a program supporting AYA cancer survivors' education and employment. Overall, I found your study well executed and reported. I have the following comments for the authors to consider:

1. page 2, line 64 "AYA with cancer describe specific identify education and career support needs, including ..." Examine the language for grammar and precision.

2. page 2, lines 68-71, you talked about AYA feel better supported when they receive formal support from education and career professionals, rather than ... While I am not surprised to hear this, I do think it is important to also acknowledge the important role of peer support, including that for their educational and career support.

3. For your integrative review, in what way it is different from or similar to the systematic review methodology? I am not questioning the use of an integrative literature review method, but one or two sentences clarifying this may help readers like me to better contextualize this method with respect to systematic review. 

4. Your study selection included those between 2011 and 2021, a one-sentence justification on the 10-year time parameter would further strengthen your paper. 

5. Your literature synthesis method can be further elaborated. What exactly did you do in terms of empirical analysis? Or, all that you did was narratively describe the findings without any "synthesis"?

6. For your study phase II, it seemed that you primarily utilized bivariate association, have the authors considered alternative analyses? e.g., regression. 

7. I enjoyed reading your study Phase III.
